# Impact of a Telemedicine Program on the Reduction in the Emission of Atmospheric Pollutants and Journeys by Road

**DOI:** 10.3390/ijerph16224366

**Published:** 2019-11-08

**Authors:** Josep Vidal-Alaball, Jordi Franch-Parella, Francesc Lopez Seguí, Francesc Garcia Cuyàs, Jacobo Mendioroz Peña

**Affiliations:** 1Health Promotion in Rural Areas Research Group, Institut Català de la Salut, 08272 Sant Fruitós de Bages, Spain; 2Unitat de Suport a la Recerca de la Catalunya Central, Fundació Institut Universitari per a la Recerca a l’Atenció Primària de Salut Jordi Gol i Gurina, 08007 Barcelona, Spain; 3Faculty of Social Sciences, Universitat de Vic-Universitat Central de Catalunya, 08242 Manresa, Spain; 4TIC Salut Social-Ministry of Health, 08005 Barcelona, Spain; 5CRES&CEXS-Pompeu Fabra University, 08005 Barcelona, Spain; 6Sant Joan de Déu Hospital, Catalan Ministry of Health, 08950 Barcelona, Spain

**Keywords:** telemedicine, carbon dioxide, air pollutants, vehicle emissions, primary care

## Abstract

This retrospective study evaluates the effect of a telemedicine program developed in the central Catalan region in lowering the environmental footprint by reducing the emission of atmospheric pollutants, thanks to a reduction in the number of hospital visits involving journeys by road. Between January 2018 and June 2019, a total of 12,322 referrals were made to telemedicine services in the primary care centers, avoiding a total of 9034 face-to-face visits. In total, the distance saved was 192,682 km, with a total travel time saving of 3779 h and a total fuel reduction of 11,754 L with an associated cost of €15,664. This represents an average reduction of 3248.3 g of carbon dioxide, 4.05 g of carbon monoxide, 4.86 g of nitric oxide and 3.2 g of sulphur dioxide. This study confirms that telemedicine reduces the environmental impact of atmospheric pollutants emitted by vehicles by reducing the number of journeys made for face-to-face visits, and thus contributing to environmental sustainability.

## 1. Introduction

The distance travelled and time spent by patients when visiting a doctor may limit patients’ access to medical care. Fortunately, telemedicine overcomes geographical barriers to healthcare, which is particularly beneficial for patients in rural communities or in places where there is a shortage of doctors or a shortage of health services [1,2,3,4].

In the central Catalan region, the Bages, Moianès and Berguedà counties have developed several telemedicine programs: the most consolidated is Teledermatology while the most innovative are Teleulcers, Teleeyelids and Teleaudiometries. Teledermatology is a service that has provided a speedy service to thousands of users, thus avoiding numerous unnecessary hospital visits [5], and representing a saving of €11.4 per face-to-face visit avoided [6]. Meanwhile, Teleulcers it is a more specialized service that, with less patient volume, has managed to improve the quality of care for people with chronic ulcers [7]. Teleeyelids and Teleaudiometries, which are in the process of being evaluated, are the most recent telemedicine programs introduced in the region.

The first three telemedicine programs all operate in the same way; the primary care physician or a nurse takes a photograph of the injury or injuries and attaches it to the patient’s electronic health record together with their clinical notes. The use of the patient’s electronic health record ensures the images are handled securely, since they do not need to be sent by email or uploaded to an external server. Hospital specialists can access the patient’s electronic health record, review the images and suggest treatment or an action plan. The primary care physician or nurse can review the instructions and telephone the patient to give them the results of the consultation. This can usually all be done in less than 5–7 working days. If the specialist has any doubts, they can ask the primary care professional to book the patient for a face-to-face consultation.

The teleaudiometry program is similar to other telemedicine programs, but it does not involve taking photographs. Patients are instead referred to a primary care center where an audiometry test is performed. This is uploaded to the patient’s electronic health record, together with specific clinical information. The patient’s otorhinolaryngologist accesses the electronic health record, reviews the audiometry test and suggests an action plan. The primary care physician reviews the instructions and telephones the patient to give them the results of the consultation as with the other programs.

Numerous studies provide evidence showing the benefits of telemedicine from the point of view of both the patient and the health system [8]. Recently some studies have analyzed the environmental impact of the savings offered by telemedicine programs due to the fact that the patient does not need to make a journey to their health center [9,10,11,12]. In the current climate of growing interest in reducing the environmental footprint of health care activities [13,14], this study evaluates the effect of a telemedicine program in lowering a procedure’s environmental footprint by reducing the emission of atmospheric pollutants due to a reduction in the number of hospital visits involving journeys by road.

## 2. Materials and Methods

The cases that were studied came from the existing telemedicine program, which includes Teledermatology, Teleulcers, Teleeyelids and Teleaudiometries in the Institut Català de la Salut’s primary health centers and tertiary referral hospitals in the Bages, Berguedà and Moianès counties; the Sant Joan de Déu Hospital (Althaia Foundation) in Manresa and the Hospital Comarcal Sant Bernabé de Berga, located in Central Catalonia. Patients from primary care centers in the Bages and Moianès counties are referred to the Hospital Sant Joan de Déu de Manresa, while patients in Berguedà are referred to the Hospital Comarcal Sant Bernabé de Berga (Figure 1). The analysis was conducted from January 2018 to June 2019.

This retrospective study uses administrative data and evaluates the impact of telemedicine services on reducing the distance travelled and the associated savings in terms of time and money, as well as the reduction in atmospheric pollutants. The savings in the distance travelled are calculated in terms of the round trip from the primary care center to the hospital.

The reduction in the emission of atmospheric pollutants and greenhouse gases is calculated by multiplying the kilometers that are not travelled by emissions per kilometer. The savings made are calculated as the difference between the cost of travelling to the referral hospital and the cost of travelling to the nearest primary care center. Finally, the time saved through the use of telemedicine is defined as the total round-trip from the primary care center to the referral hospital. Google Maps was used to calculate both the distances travelled and the time saved on the trip via the existing road network. The “fastest route with the usual traffic” search option was used to make the calculations, and involved an equal number of diesel and petrol cars. The emission of atmospheric pollutants per km is shown in Table 1.

The telemedicine service was considered to have replaced a face-to-face consultation when no face-to-face visit occurred in relation to the same type of specialist in the three months following the telemedicine consultation. Microsoft Excel and R 3.6.1 were used for data processing and quantitative analysis.

## 3. Results

As noted in Table 2, between January 2018 and June 2019, a total of 12,322 referrals were made to telemedicine services in the primary care centers which are within the catchment areas of the 2 referral hospitals, avoiding a total of 9034 face-to-face visits. The remaining 3288 telemedicine consultations were referred to a specialist for a face-to-face consultation. Thus, the percentage of face-to-face visits prevented through the use of telemedicine was 73.3%. The most widely used telemedicine service was Teledermatology, with 9352 cases and a reduction in face-to-face visits of 69.66%, followed by Teleaudiometries (2465 cases and a 86.10% reduction in visits), Teleeyelids (350 cases and a 74.6% reduction in visits) and Teleulcers (155 cases and 87.8% reduction in visits).

Table 3 shows the distance of saved visits due to the telemedicine program and time and travel cost savings in terms of unused fuel. The average distance of a round trip between a primary care center and the referral hospital was 21.3 km, with an average travel time savings of 25 min. In total, the distance saved was 192,682 km, with a total travel time savings of 3779 h and a total fuel reduction of 11,754 L with an associated cost of €15,664.

A breakdown of the reduction in the emissions of pollutants is shown in Table 4. Total reduction in pollutants is 29.384 tonnes of carbon dioxide, 36.61 kg of carbon monoxide, 43.93 kg of nitric oxide and 28.9 kg of sulphur dioxide. This represents an average reduction of 3248.3 g of carbon dioxide, 4.05 g of carbon monoxide, 4.86 g of nitric oxide and 3.2 g of sulphur dioxide.

## 4. Discussion

In this study, the results of a fully functioning and fully developed telemedicine program comprising four different services, Teledermatology, Teleulcers, Teleeyelids and Teleaudiometries, have been analyzed in terms of journeys to hospitals prevented, time saved, reductions in fuel consumption and reductions in the emission of pollutant and greenhouse gases. The rural and dispersed nature of the area involved increases the effectiveness of telemedicine. The three counties featured in the study have a total population of 228,622, with 66.65% living in rural areas, with an urban area defined as having a population greater than 20,000. From the patient’s point of view, they avoid an unnecessary journey to hospital, save on the journey time and reduce their fuel consumption while receiving the same standard of care. From the environmental point of view, there is a reduction in the emission of pollutants and greenhouse gases. 

The results are consistent with those from 2016 [6], involving the same geographical area, and relating specifically to the Teledermatology service. Of a total of 5606 consultations, 80.3% were dealt with telematically, while the remaining 19.7% were referred to a specialist for a face-to-face consultation. In 2016, Teledermatology reduced journeys by a total of 99,368 km. This represented 1928 h of travel time saved. CO_2_ emissions were reduced by 15.2 tonnes. The study included a cost-saving analysis that concluded that using teledermatology instead of face-to-face dermatology consultations could save up to €11.4 per patient visited. Other studies also suggest that the reduction in visits as a result of telemedicine services and the consequent reduction of pollutant emissions is significant [15,16]. A recent study conducted by the vascular surgery division of the Henry Ford hospital in Detroit between October 2015 and November 2017 [17] analyzed the impact of telemedicine on 87 patients. The results showed that the average reduction in pollutants per consultation was 1118 g of carbon dioxide, 294 g of carbon monoxide, 21.6 g of nitric oxide and 32.3 g of other volatile organic compounds. The average distance saved on a round trip was 50.2 km, with an average duration of 39 min. The total reduction in fuel consumption was 734 L with an associated cost of $622. This includes $2.50 for parking. Another study by the University of Kansas Medical Center, involving 132 patients and a total of 257 consultations shows an average saving per journey of $86.13 [18], while in another study by the University of Kentucky, the distance and time saved per visit as a result of telemedicine was 102 km and 66.8 min, respectively [19].

The present study has certain limitations. A prospective analysis would be able to obtain more detailed and individualized information, including data related to the loss of working hours and salary, the degree of driving-related stress, waiting time and additional costs such as parking. None of these costs are included in the current study. In addition, this study does not take into account factors that increase the cost of telemedicine such as the use of platforms or the need for equipment connected to the internet.

For the purposes of the study, the type of transport used has been taken to be a private vehicle with average emissions. The use of public transport or emissions that deviate from the average (whether due to lower emissions through the use of hybrid or electric cars, or higher emissions through the use of outdated and obsolete cars) have not been taken into account. However, public transport is not always easily available in rural areas.

Future studies ought to look at the monetary value of the pollutants which were not emitted, because telemedicine, in addition to saving on direct costs, can contribute to environmental sustainability.

## 5. Conclusions

This study confirms that telemedicine reduces the environmental impact of atmospheric pollutants emitted by vehicles by reducing the number of journeys made for face-to-face visits to see a specialist. The distance saved results in time saved and reductions in fuel and pollutants. Further studies ought to include more information on alternative face-to-face models, wasted working hours and the associated impact on earnings, as well as a cost-effectiveness analysis of face-to-face visits compared to telemedicine. The expansion of telemedicine programs ought to be considered an option as part of a global strategy to reduce the emission of atmospheric pollutants. 

## Figures and Tables

**Figure 1 ijerph-16-04366-f001:**
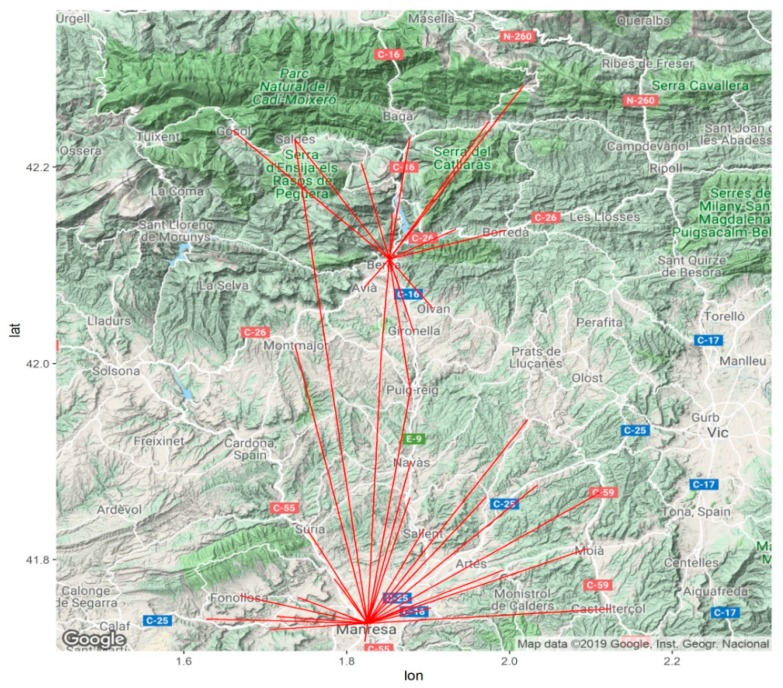
Map showing the primary care centers and their associated referral hospitals in Manresa and Berga.

**Table 1 ijerph-16-04366-t001:** Emission of pollutants per km. Source: Mobility and emissions. Extra-urban cycle (gasoline and diesel fuel). Available at: http://mobilitat.gencat.cat/ca/detalls/Article/mobilitat_emissions (Date of access: 20 September 2019).

Pollutant	Formula	Emissions per km (g)
Carbon dioxide	CO_2_	152.5
Carbon monoxide	CO	0.19
Nitric Oxide	NO_x_	0.228
Sulphur dioxide	SO_2_	0.15

**Table 2 ijerph-16-04366-t002:** Face-to-face visits and visits prevented through telemedicine.

	Teledermatology	Teleaudiometries	Teleeyelids	Teleulcers	Total
Origin of Visits	9352	2465	350	155	12,322
V. Referred	2837	343	89	19	3288
V. Prevented	6515	2122	261	136	9034
% V. Prevented	69.66%	86.10%	74.6%	87.8%	73.3%

**Table 3 ijerph-16-04366-t003:** Reduction in journeys by distance, time, fuel and cost. Cost of fuel available at: https://www.dieselgasolina.com (Accessed on: 17 September 2019).

Kms	
Average journey saved in km (return)	21.3 km
Total journeys saved (return)	192,682 km
Time	
Average journey saved in minutes (return)	25 min
Total journeys saved in hours (return)	3779 h
Total reduction in petrol consumption	11,754 L
Cost of fuel saved	15,664 euros

**Table 4 ijerph-16-04366-t004:** Reduction in the emissions of pollutant gases.

Carbon Dioxide	
Average (return) journey	3248.3 g
Total journeys	29.384 t
Carbon monoxide	
Average journey	4.05 g
Total journeys	36.61 kg
Nitric oxide	
Average journey	4.86 g
Total journeys	43.93 kg
Sulphur dioxide	
Average journey	3.2 g
Total journeys	28.9 kg

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
