# Peer review of "Impact of a Telemedicine Program on the Reduction in the Emission of Atmospheric Pollutants and Journeys by Road"

_ijerph, 2019, doi:10.3390/ijerph16224366_

Round 1

Reviewer 1 Report

The reviewed article entitled "Impact of a telemedicine program on the reduction in the emission of atmospheric pollutants and journeys by road" concerns an interesting and very current problem related to the need to reduce atmospheric pollution emitted by human.

The authors evaluate the effect of a telemedicine program developed in the central Catalan region in lowering the environmental footprint by reducing the emission of atmospheric pollutants due to a reduction in the number of hospital visits involving journeys by cars.

The results obtained are interesting and refer to the current results of research on similar issues conducted in the world.

The article was written clearly and correctly in scientific terms. The aim of the work was set correctly. The research methods used are adequate to the set goal.

Comments

The description of the tables should be above the tables. Table 3 - the information contained in the first column, first row (Kms) is incomprehensible. Table 4 - unit abbreviations should be corrected (Kg per kg, Tn per t)

4th line 166 - there should be a period at the end of the sentence (there is a comma).

Author Response

Many thanks for your comments that will help us improving our manuscript.

Comments:

The description of the tables should be above the tables.

We have addressed this

Table 3 - the information contained in the first column, first row (Kms) is incomprehensible.

The table shows the distance of saved visits due to the telemedicine services and time and travel cost savings in terms of unused fuel. We have clarified this in the table and in the text.

Table 4 - unit abbreviations should be corrected (Kg per kg, Tn per t)

We have addressed this

4th line 166 - there should be a period at the end of the sentence (there is a comma).

We have removed the comma, but we are not sure it is the one you mention.

Reviewer 2 Report

Referee Report on “Impact of a telemedicine program on the reduction in the emission of atmospheric pollutants and journeys by road”.

Manuscript number Ijerph-620883

This retrospective study evaluates the effect of a telemedicine program developed in the central Catalan region in lowering the environmental footprint by reducing the emission of atmospheric pollutants thanks to a reduction in the number of hospital visits involving journeys by road. In total, the distance saved was 192,682 km, with a total travel time savings of 3,779 hours and a total fuel reduction of 11,754 liters with an associated cost of €15,664.  

It is an interesting paper. The author confirms that telemedicine reduces the environmental impact of atmospheric pollutants emitted by vehicles, reducing the number of journeys made for face-to-face visits and can contribute to environmental sustainability. However, I have the following specific concerns.

Major Concerns and Comments:

The authors propose environmental improvements and economic benefits from telemedicine. However, the lack of discussion and description of the cost involved also reduces the contribution of this paper. It is recommended that authors be able to conduct cost-benefit analysis, especially if the government and consumers are generating more medical burdens due to the use of telemedicine.

The authors suggest that similar conclusions are obtained compared to previous studies, including reductions in environmental pollution, time and cost savings. So, I don't know “what the main contribution of this research is?” Is it just a country different and gets similar results? How to highlight their contribution is still to be discussed by the author.

Minor Comments:

In line 164, “,” should be ”.”.

Final Evaluation:

For the above reasons, I believe that the current situation in this article is not suitable for publication in this journal.

Author Response

Many thanks for your comments that will help us improving our manuscript.

Major Concerns and Comments:

The authors propose environmental improvements and economic benefits from telemedicine. However, the lack of discussion and description of the cost involved also reduces the contribution of this paper. It is recommended that authors be able to conduct cost-benefit analysis, especially if the government and consumers are generating more medical burdens due to the use of telemedicine.

Yes, we fully agree with your comments. In fact in 2016 we conducted a cost-saving analysis in the same region but just evaluating the teledermatology service, we concluded that using teledermatology instead of face-to-face dermatology consultations could save up to 11.4 € per patient visited. We have incorporated this in the discussion. For the future we are planning to undergo a cost-benefit analysis of all the telemedicine services.

The authors suggest that similar conclusions are obtained compared to previous studies, including reductions in environmental pollution, time and cost savings. So, I don't know “what the main contribution of this research is?” Is it just a country different and gets similar results? How to highlight their contribution is still to be discussed by the author.

Yes, we also agree with you remarks. However, our study is different because we evaluate a whole telemedicine program, comprising 4 different services: Teledermatology, Teleulcers, Teleeyelids and Teleaudiometries. So far, nobody has evaluated the impact of very innovative services such as teleeyelids, teleaudiometries or teleulcers. Moreover, we don’t just evaluate a pilot program, we are evaluating a fully developed, fully functioning and clinically fully integrated program and this has not been done by the other studies.

We have tried to empathize this in the text,

Minor Comments:

In line 164, “,” should be ”.”.

We have addressed this

Round 2

Reviewer 2 Report

No comments.